# Failure Prediction and Replacement Strategies for Smart Electricity Meters Based on Field Failure Observation [note 1]

**DOI:** 10.3390/s22249804

**Published:** 2022-12-14

**Authors:** Xianguang Dong, Zhen Jing, Yanjie Dai, Pingxin Wang, Zhen Chen

**Affiliations:** 1State Grid Shandong Electric Power Company Metering Service Marketing Center, Jinan 250001, China; 2Institute of Sensor and Reliability Engineering, Harbin University of Science and Technology, Harbin 150080, China

**Keywords:** failure number prediction, electricity meter, Weibull distribution, replacement strategies, Bayesian

## Abstract

It is helpful to have a replacement strategy by predicting the number of failures of in-service electricity meters. This paper presents a failure number prediction method for smart electricity meters based on on-site fault data. The prediction model was constructed by combining Weibull distribution with odds ratios, then the distribution parameters, failure prediction number, and confidence intervals of prediction number were calculated. A strategy of meter replacement and reserve were developed according to the prediction results. To avoid the uncertainty of prediction results due to the small amount of field data information, a Bayesian failure number prediction method was developed. The research results have value for making operation plans and reserve strategies for electricity meters.

## 1. Introduction

More than 0.5 billion electricity meters have been installed in China since smart electricity meters started being implemented in 2009. The smart meters were replaced when the operation period reached eight years. Smart electricity meters are seriously affected by various factors, such as temperature, humidity, thunder and lightning, power system fluctuation, and electromagnetic interference, and the failure risk increases with long-term operation. With the increased number of installed electricity meters, failures also grew gradually. Failure number prediction is necessary for decision-making on spare parts inventory. In addition, the requirement for in-service life of smart electricity meters was enhanced by the state grid from 8 years to 16 years in 2020. The service period of installed meters will reach 8 years soon. Whether or when to replace the 8-year meters is a challenging problem. In this paper, we would like to investigate this problem and provide the technical basis for the phased rotation of smart electricity meters.

Ref. [1] combined the k-means clustering and bidirectional long- and short-term neural networks to predict the performance degradation trend of built-in electricity meter relays, and then evaluated their reliability. In Ref. [2], the mean impact value (MIV) algorithm was combined to predict the reliable life of the contactor through an adaptive BP neural network model. Ref. [3] evaluated the reliability of smart electricity meters according to the Bayesian method using accelerated degradation data. Ref. [4] analyzed and predicted the reliability of each module of the electricity meters based on the VC++ platform. Ref. [5] established the LSTM model based on Harris hawks optimization and predicted its reliable life by measuring the contact resistance value. Ref. [6] used the least square method to calculate Weibull distribution parameters and then evaluated the reliability of batch electricity meters. Ref. [7] examined the fault data under external stress conditions to predict the remaining life of the smart electricity meters in operation. Existing studies analyzed electricity meter reliability based on laboratory tests, and a few investigations focused on historical fault data. In addition, previous research focused on reliability estimation and life prediction rather than failure number prediction.

Existing studies showed that the Bayes method was effective in reliability assessment for small sample products. Ref. [8] proposed a machine learning technology to predict rainfall, which integrated Bayes theory and achieved good prediction results. Ref. [9] introduced Bayes theory into the neural network model to optimize the estimation of relevant parameters and improve the accuracy, adaptability, and generalization ability of SOC estimation of batteries with different chemical components. Ref. [10] used the Bayesian method to update the previous estimation of users’ consumption habits by using the actual power load information, predicting the hourly load situation. It effectively evaluated the household energy use pattern and users’ consumption habits. Ref. [11] used several different machine learning technologies, among which Bayesian methods were integrated to predict metabolic sites (SOMs) for xenogeneic detection, which solved the most critical problems in the first stage of metabolic prediction. Ref. [12] used the naïve Bayes data-driven model to determine the formation conditions of pure gas hydrate and mixed gas hydrate, which was applied to predict the formation conditions of gas hydrate, effectively reducing the uncertainty of prediction results. Although the Bayes method has a good effect, there was no research on the prediction of smart electricity meter failure numbers that used the Bayes method. Electricity meters have the characteristics of large quantity and comprehensive coverage. For example, there were more than 50 million electricity meters in a province. Similar products and historical data produced a large quantity of failure data of electricity meters, which reflected the reliability of electricity meters. To make use of multisource reliability information, the Bayes method was adopted to expand the information capacity in the form of previous data and improve the accuracy of prediction evaluation [13].

Ref. [14] reviewed the latest progress in the application for smart electricity meter data and discussed the directions of using smart electricity meter data for analysis. Ref. [15] calculated a distribution system reliability index according to smart electricity meter data. Ref. [16] used smart meter data and household features data to seek the most appropriate methods of energy consumption prediction. Using the cross-industry standard process for data mining (CRISP-DM) method, support vector machines, random forest regression and neural networks methods, prediction experiments were performed with household feature data and past consumption data of over 470 smart meters that gathered data for three years. The results help utilities to offer better contracts to new households and to manage their smart grid infrastructure based on the forecast demand. In addition, Refs. [17,18] used neural networks and hierarchical probabilistic forecasts methods to predict energy consumption. Ref. [19] established a nonlinear Wiener process prediction model to calculate the reliable life of the contactor. Ref. [20] collected the degradation data before periodic censoring to analyze and calculate product reliability. Ref. [21] used the time series algorithm to predict the monthly failure number of electricity meters. Ref. [22] presented an approach of ε-support vector regression (ε-SVR) for predicting the remaining useful life of bearings. Most of the above prediction methods used neural networks, support vector machines, random forest and time series algorithms. However, most of the above prediction methods require a large quantity of data and are cumbersome to construct. This paper skillfully constructs a mathematical prediction model according to the field data, which is a simple and efficient way to solve the problem. To make the prediction result more accurate, this paper introduces the odds ratio to study the failure number prediction in the future interval under the Weibull distribution.

The arrangement of this paper is as follows: Section 2 is the prediction of failure numbers in the future, Section 3 is a replacement strategy for smart electricity meters, and Section 4 is a case study. The effectiveness of this method was verified by engineering examples. The research results provide technical support for the operation and maintenance of electricity meters and improving life cycle management.

## 2. Prediction of Failure Number

### 2.1. Smart Electricity Meters

The smart electricity meter is essential equipment for smart grid data collection, and undertakes the tasks of raw power data collection, measurement and transmission. It is the basis of information integration, analysis and optimization and information presentation. In addition to the metering function of primary electricity consumption of traditional electricity meters, smart electricity meters also have intelligent functions, such as bidirectional multi-rate metering function, user-end control function, bidirectional data communication function of various data transmission modes, anti-electric theft function and so on, in order to adapt to the use of smart grids and new energy [16]. Smart electricity meters are essential equipment for power grid companies. Their operation state affects the stability and power supply security of the whole power grid system, and it is a bridge to realize the communication between enterprises and users. With the wide application of smart electricity meters, a large number of fault data are generated in the field, which will be of great help in the prediction of failure number of electricity meters and the formulation of the replacement strategy of electricity meters in the future.

### 2.2. Field Fault Data

The field fault data structure is shown in Figure 1. After the electricity meter was put into operation, the fault data of the electricity meters were collected in the specified time $t_{s}$. The time fault data after the specified time $t_{s}$ could not be obtained. Let the number of electricity meters be N and be put into use in a batch, and G electricity meters failed within the observation time from 0 to $t_{s}$*_._* Failure can be detected no matter how small the observation time from 0 to $t_{s}$, but to predict better, we need fault data with longer observation time. The failure time was recorded as $t_{1}$, $t_{2}$,…,$t_{G}$*,* respectively. If the time is set as $t_{e}$, the number of faults within the future prediction time interval △t from $t_{s}$ to $t_{e}$ is H. To maintain the regular operation of the equipment, we need to reserve H electricity meters. After setting the forecast time in the future, the number of electricity meters was still H in this batch, so they obeyed G+H+K=N, and the probability of $\left(G,\;H,\;K\right)$ corresponding occurrence was g*,*
h*, and* k, respectively, where g+h+k=1. In addition, the future prediction time interval △t from $t_{s}$ to $t_{e}$ is set according to the actual demand of the power company. Under normal circumstances, the power company will bid for the meter every year, so we only need to predict the number of meter failure in the following year. As such, the prediction time interval △t from $t_{s}$ to time $t_{e}$ is generally one year, where $△t=t_{e}-t_{s}$.

### 2.3. Life Distribution of Smart Electricity Meters

It is known that the life of an electricity meter follows the Weibull distribution [23,24]. Let the energy of an electricity meter be expressed as a random variable obeying the Weibull distribution with time t, then the probability density function and cumulative failure probability, respectively at time t is:(1)$$f(t)=\frac{m}{\eta }{\left(\frac{t}{\eta }\right)}^{m-1}e^{-{\left(\frac{t}{\eta }\right)}^{m}}$$
(2)$$F(t)=1-e^{-{\left(\frac{t}{\eta }\right)}^{m}}$$
where η is the scale parameter and m the shape parameter.

The Weibull distribution parameters were estimated as follows:(3)$$m^{-1}=\frac{\sum_{i=1}^{G}t_{{}_{i}}^{m}\ln t_{i}+(N-G)t_{s}{}^{m}\ln t_{s}}{\sum_{i=1}^{G}t_{i}^{m}+(N-G)t_{s}{}^{m}}-\frac{1}{G}\sum_{i=1}^{G}\ln t_{i}$$
(4)$$\eta ={\left(\frac{\sum_{i=1}^{G}(t_{i}{)}^{m}+(N-G)t_{s}{}^{m}}{G}\right)}^{m^{-1}}$$
where N was the total number of installs in batches of electricity meters, G was the number of failures within the observation time, the corresponding time of each failure was $t_{i}$(i = 1, 2, …, G), and $t_{s}$ was the observation time.

The above two equations were maximum likelihood estimates of Weibull parameters. The two parameters were also necessary parameters of the following failure number prediction equation and failure number interval prediction equation, so the estimation results of the two parameters played a crucial role in predicting failure number and failure number interval.

### 2.4. Failure Number Predictor

Combined with the engineering example data, the historical failure number G can be obtained within the observation time, while the predicted failure number H is unknown. Therefore, the cumulative fault probability g and h can be calculated first. Then, the point estimate of the future interval failure number H can be obtained by the cumulative fault probability from time $t_{s}$ to time $t_{e}$ and the sample size.

Then, under the Weibull distribution, the cumulative failure probability of products at $t_{s}$ and $t_{e}{}_{}$ was:(5)$$g=F(t_{s})=1-e^{-{\left(\frac{t_{s}}{\eta }\right)}^{m}}$$
(6)$$F(t_{e})=1-e^{-{\left(\frac{t_{e}}{\eta }\right)}^{m}}$$

According to Equation (6), we can get:(7)$$k=1-F(t_{e})=e^{-{\left(\frac{t_{e}}{\eta }\right)}^{m}}$$

According to Equations (5) and (7) above, the cumulative fault probability within the interval from time $t_{s}$ to time $t_{e}{}_{}$ can be obtained:(8)$$\hat{h}=1-g-k$$

Then, the point estimation of H was approximate as follows:(9)$$\hat{h}=e^{-{\left(\frac{t_{s}}{\eta }\right)}^{m}}-e^{-{\left(\frac{t_{e}}{\eta }\right)}^{m}}$$

Therefore, the point estimation of the failure number H within the future interval from time $t_{s}$ to the time $t_{e}$ under Weibull distribution can be approximated as:(10)$$\hat{H}=N\hat{h}$$

In the above equation, N was the total installed amount of a batch of electricity meters. The equation derived here was used in failure number prediction.

### 2.5. Construction of the Odds Ratio

The traditional method obtained the prediction results by multiplying the sample size and the cumulative failure probability. It failed to give the prediction distribution of the failure number, so the prediction risk was high. If the odds ratio is constructed using the ratio of the cumulative fault probability in the observation interval to the future period, and the point estimation and confidence limit of the odds ratio are calculated, then the inverse operation of the odds ratio can be used.

To construct the odds ratio, $\left(g_{o},h_{o},k_{o}\right)$ was set as the observed value of the random variable (G,H,K) in Section 2.1, and the corresponding occurrence probability was (g,h,k), then (G,H,K) followed a three-term distribution.
(11)$$p(g_{o},h_{o},k_{o})=\frac{N!}{g_{o}^{2}!h_{o}^{2}!k_{o}^{2}!}g^{g_{o}}h^{h_{o}}k^{k_{o}}$$
where G+H+K=N and  g+h+k=1.

The odds ratio was:(12)β=g/h
where β was the odds ratio, then its point estimation was approximate:(13)$$\hat{\beta }=\hat{g}/\hat{h}$$

When the confidence degree γ was given, the confidence interval of the parameter β was:(14)$$\left\{\begin{array}{l} \beta_{L}=\frac{G}{H+1}F_{\left(1-\gamma \right)/2}(2G,2H+1)\\ \beta_{U}=\frac{G}{H+1}F_{(1+\gamma )/2}(2G+2,2H) \end{array}\right.$$
where $F_{\left(1-\gamma \right)/2}(2G,2H+1)$ and $F_{(1+\gamma )/2}(2G+2,2H)$ were the quantile of the f-distribution with the confidence of $\left(1-\gamma \right)/2$ and $\left(1+\gamma \right)/2$. When the random variables G and H are known, the confidence limits of the odds ratio can be obtained. This equation was a necessary condition for constructing the following prediction intervals of the failure number equation.

### 2.6. Prediction Intervals of Failure Number

To solve the problem of failure number interval prediction under Weibull distribution and avoid the high risk of traditional prediction method, the prediction equation of failure number interval was constructed.

According to Equation (13), the point estimation of the odds ratio under the Weibull distribution is approximate:(15)$$\hat{\beta }=\hat{g}/\hat{h}=\frac{1-e^{-{\left(\frac{t_{s}}{\eta }\right)}^{m}}}{e^{-{\left(\frac{t_{s}}{\eta }\right)}^{m}}-e^{-{\left(\frac{t_{e}}{\eta }\right)}^{m}}}$$

Given the confidence degree γ, Equation (13) can be used:(16)$$\left\{\begin{array}{l} \hat{\beta }\geq \frac{G}{H_{L}+1}F_{(1-\gamma )/2}(2G,2H_{L}+2)\\ \hat{\beta }\leq \frac{G+1}{H_{U}}F_{(1+\gamma )/2}(2G+2,2H_{U}) \end{array}\right.$$

The number of failure G in the observation time was known, and the point estimation of the odds ratio $\hat{\beta }$ was obtained according to Equation (15). It can be understood that both ends of the inequality in Equation (16) are functions that take the number H of failure in the future period as the variable and are monotonically decreasing. Therefore, let $H_{L}$ be the minimum value for the left inequality to hold, i.e., let $H_{L}$ be the lower limit of H and let $H_{U}$ be the maximum value for the correct imbalance to have, i.e., let $H_{U}$ be the upper limit of H, where the confidence γ in this paper is 0.9. The confidence limits of failure can be obtained by iteration.

The equation derived in this subsection was applied to the prediction of ntervals of failure number.

### 2.7. Bayes Estimate

For Weibull life distribution, where, $\lambda =\eta^{-m}$, η was the scale parameter, m was the shape parameter. Combining Equations (1) and (2), the probability density function was updated:(17)$$f(t)=\lambda mt^{m-1}e^{\left(-\lambda t^{m}\right)}$$

The accumulated failure function was:(18)$$F(t)=1-e^{\left(-\lambda t^{m}\right)}$$
the parameter λ was:(19)$$\lambda =-\ln R/t_{R}^{m}$$
where R is the degree of reliability.

Suppose the parameters obey the gamma distribution $\lambda ∼\;Ga\left(a,b\right)$:(20)$$\pi (\lambda )=f(\lambda /a,b)=\frac{b^{a}\lambda^{a-1}}{\Gamma (a)}e^{\left(-\lambda b\right)}$$

The reliable life of electricity meters has been increased from 8 years to 16 years, which equates to for 2920 days and 5840 days, respectively. Therefore, the range of the parameter λ was determined as $\left[-\ln R/5840^{m},-\ln R/2920^{m}\right]$, which represented the value of the failure rate of the electricity meters.

According to the 3σ criterion, the value range of parameter λ was about 6σ:(21)$$6\sigma =(\ln R/5840^{m}-\ln R/2920^{m})$$
and the variance of failure rate was:(22)$$\sigma^{2}={\left(\ln R/5840^{m}-\ln R/2920^{m}\right)}^{2}/36$$

The mean value was:(23)$$v_{a}=(-\ln R/5840^{m}-\ln R/2920^{m})/2$$

The moment estimation method was used to calculate the previous distribution parameters a and b:(24)$$\left\{\begin{array}{l} a/b=E(\lambda )=v_{a}=(-\ln R/5840^{m}-\ln R/2920^{m})/2\\ a/b^{2}=D(\lambda )={\left(\ln R/5840^{m}-\ln R/2920^{m}\right)}^{2}/36 \end{array}\right.$$

The previous distribution parameters can be obtained as:(25)$$\left\{\begin{array}{l} a=\frac{9{\left(-\ln R/5840^{m}-\ln R/2920^{m}\right)}^{2}}{{\left(\ln R/5840^{m}-\ln R/2920^{m}\right)}^{2}}\\ b=\frac{18(-\ln R/5840^{m}-\ln R/2920^{m})}{{\left(\ln R/5840^{m}-\ln R/2920^{m}\right)}^{2}} \end{array}\right.$$

After obtaining the current test observation data, the posterior distribution of Weibull distribution of the meter life model was:(26)$$p(\lambda ,m/t)=\frac{\pi (\lambda ,m)L(t/\lambda ,m)}{\iint_{\lambda ,m}\pi (\lambda ,m)L(t/\lambda ,m)d_{\lambda }d_{m}}$$
where π(λ,m) was the previous distribution of the two parameters, which can be extracted from the on-site fault data information, and L(t/λ,m) was the likelihood function of the life observation value:(27)$$\begin{array}{l} L(t/\lambda ,m)=\prod_{i=1}^{G}f(t)=\prod_{i=1}^{G}(\lambda mt^{m-1}e^{\left(-\lambda t^{m}\right)})\\ =\lambda^{G}m^{G}\prod_{i=1}^{G}t_{i}{}^{m-1}e^{\left(-\lambda \sum_{i=1}^{G}t_{i}^{m}\right)} \end{array}$$

If the parameter $m_{b}$ was initially estimated by the classical method, the posterior density function distribution could be calculated as [25]:(28)$$p(\lambda /t)=\frac{\pi (\lambda /a,b)L(t/\lambda )}{\int_{0}^{\infty }\pi (\lambda /a,b)L(t/\lambda )d_{\lambda }}$$

Then, the Bayes estimate of the parameter λ was:(29)$${\hat{\lambda }}_{b}=\frac{G+a}{b+\sum_{i=1}^{G}t_{i}{}^{m}}$$
where a and b were the previous distribution parameters, the meter failure time was $t_{i}$ (i = 1,2,…,G), and G was the number of failures. The corresponding Bayesian cumulative failure rate was:(30)$${\hat{g}}_{b}=1-e^{\left(-{\hat{\lambda }}_{b}t_{s}{}^{m}\right)}$$

Then, the cumulative failure rate from $t_{s}$ to $t_{e}$ was estimated as:(31)$${\hat{h}}_{b}=e^{\left(-{\hat{\lambda }}_{b}t_{s}{}^{m}\right)}-e^{\left(-{\hat{\lambda }}_{b}t_{e}{}^{m}\right)}$$

Combined with Equation (10), the Bayes point estimation of the failure number of the electricity meters can be obtained:(32)$${\hat{H}}_{b}=N(e^{\left(-{\hat{\lambda }}_{b}t_{s}{}^{m}\right)}-e^{\left(-{\hat{\lambda }}_{b}t_{e}{}^{m}\right)})$$

Combined with Equation (14), the Bayes estimated odds ratio was:(33)$${\hat{\beta }}_{b}={\hat{g}}_{b}/{\hat{h}}_{b}=\frac{1-e^{\left(-{\hat{\lambda }}_{b}t_{s}{}^{m}\right)}}{e^{\left(-{\hat{\lambda }}_{b}t_{s}{}^{m}\right)}-e^{\left(-{\hat{\lambda }}_{b}t_{e}{}^{m}\right)}}$$

According to Equation (15), the Bayes confidence limits of the failure number of meters can be obtained:(34)$$\left\{\begin{array}{l} {\hat{\beta }}_{b}\geq \frac{G}{H_{Lb}+1}F_{(1-\gamma )/2}(2G,2H_{Lb}+2)\\ {\hat{\beta }}_{b}\leq \frac{G+1}{H_{Ub}}F_{(1+\gamma )/2}(2G+2,2H_{Ub}) \end{array}\right.$$
where $F_{\left(1-\gamma \right)/2}(2G,2H_{Lb}+1)$ and $F_{(1+\gamma )/2}(2G+2,2H_{Ub})$ were the quantile of the f-distribution with confidence of $\left(1-\gamma \right)/2$ and $\left(1+\gamma \right)/2$. The number of failure G in observation time was known, and the point estimation of the odds ratio ${\hat{\beta }}_{b}$ was obtained according to Equation (33). It can be understood that both ends of the inequality in Equation (34) were functions that take the number H of failures in the future as the variable and were monotonically decreasing. Therefore, let $H_{Lb}$ be the minimum value for the left inequality to hold, i.e., let $H_{Lb}$ be the lower limit of ${\hat{H}}_{b}$, and let $H_{Ub}$ be the maximum value for the correct imbalance to have, i.e., let $H_{Ub}$ be the upper limit of H, where the confidence γ in this paper was 0.9. The confidence limits of failure can be obtained by iteration.

### 2.8. Prediction Precision Analysis

The relative dispersion degree of the prediction results of the number of failure in the future time interval can be expressed by the range coefficient E of the number of failures H, which reflected the relative dispersion ratio of the number of failures H in the future time interval near its expected value.
(35)$$E=\frac{H_{U}-H_{L}}{\hat{H}}$$

The value of E was to 1. The closer the failure number H was to its expected value, the lower the relative dispersion degree was.

The corresponding Bayesian dispersion degree was:(36)$$E_{b}=\frac{H_{Ub}-H_{Lb}}{{\hat{H}}_{b}}$$

## 3. Replacement Strategies for Smart Electricity Meters

The decision-making procedure for electricity meter quantity is shown in Figure 2. Firstly, the failure number of electricity meters in the future was predicted based on the on-site failure data. Then, the number of spare parts can be expected based on the failure number prediction of smart electricity meters. In addition, the accumulated failure number of meters, the failure rate, and the proportion of failure meters to all the electricity meters in the same batch can be predicted. At the same time, the operation time of the meter after the prediction was obtained by adding the running time of the field data to the future prediction interval. The batch electricity meters will be rotated if the failure proportion exceeds the threshold $R_{D,rotate}$ or the operation time is longer than the specified time $T_{D,rotate}$.

### 3.1. Decision-Making on Spare Parts Quantity of Smart Electricity Meters

The number of installed meters in the $l^{th}$ month was $z_{l}$ (l = 1,2,…,12). After T years of operation, $r_{l}$ failed meters were observed. The operation time was $t_{l,j}$ (j = 1, 2, …, $r_{l}$), $s_{l}$ meters continued to work, the operation time was $t_{l,q}$ (q = $r_{l}$
*+* 1, $r_{l}$ + 2, …, $z_{l}$), and the total number of installed meters in a year was $Z=sum\;(z_{l})$.

The failure number of meters installed in the $l^{th}$ month in the next year $r_{l}{}_{,YF}$ can be predicted. According to Equation (10), the total failed meters in the next year is:(37)$$r_{YF}=sum\left(r_{l}{}_{,YF}\right)$$

According to Equation (28), the Bayesian estimate of the total failure number in the next year was estimated:(38)$$r_{YFb}=sum_{b}\left(r_{l}{}_{,YF}\right)$$

Therefore, the decision that $r_{YF}$ smart electricity meters need to be reserved can be made for the following year.

### 3.2. Decision-Making on Rotation Time of Batch Smart Electricity Meters

The cumulated failure number of meters in the next year was:(39)$$r_{Accu}=sum\left(r_{l}\right)+r_{YF}$$

The corresponding Bayesian estimate of the accumulated failure number of electricity meters in the next year was:(40)$$r_{Accu}=sum\left(r_{l}\right)+r_{YF}$$

The proportion of failed meters to all meters $R_{F/ALL}$ was computed:(41)$$R_{F/ALL}=r_{Accu}/Z$$

The Bayesian estimation was:(42)$$R_{F/ALLb}=r_{Accub}/Z$$

Decision making:

If the failure proportion exceeded the threshold $R_{D,rotate}$,
(43)$$R_{F/ALL}\geq R_{D,rotate}$$
or
(44)$$R_{F/ALLb}\geq R_{D,rotate}$$
or the operation time was longer than the specified time $T_{D,rotate}$:(45)$$t_{Operation}\geq T_{D,rotate}$$

The batch of electricity meters was rotated. The quantity of rotated smart electricity meters $N_{rotate}$ was:(46)$$N_{rotate}=Z-r_{Accu}$$

The Bayes quantity of rotated smart electricity meters was:(47)$$N_{rotateb}=Z-r_{Accub}$$

The equations derived in this subsection are applied to the rotation of electricity meters.

## 4. Example Analysis

### 4.1. Data Source

In August 2017, 578 electricity meters were installed, and the field operation data between 2017 and February 2022 were collected. The failure time and cumulative failure rate of 35 failed electricity meters by the end of 2019 are shown in Table 1.

Combined with the data in the table above, scale parameters η and shape parameters m can be calculated according to Equations (3) and (4). According to Equations (9) and (10), the failure rate h and the failure number H of each future prediction time interval △t can be obtained. The confidence interval of H can be calculated by Equation (16), and the relative dispersion degree of failure number H can be calculated according to Equation (35). The future prediction time interval △t in the table below represented 365 days from the end of 2019 to the end of 2020, 730 days from the end of 2019 to the end of 2021, and 790 days from the end of 2019 to the end of February 2022, respectively. The responding calculation results are shown in Table 2 and Table 3 below.

The actual failure of electricity meters was 9, 23, and 27 in the following 365 days, 730 days, and 790 days, respectively.

**Table 3 sensors-22-09804-t003:** Prediction interval and relative dispersion ratio.

△t	Bilateral Interval	E
365	[7.295, 19.48]	0.9151
730	[16.66, 35.34]	0.7193
790	[18.18, 37.86]	0.7027

where △t was the future prediction time interval, $△t=t_{e}-t_{s}$, and the unit was days.

As seen from the above table, among the three future time intervals, when the future time interval was 365 days, the relative dispersion ratio was the largest and close to 1. The relative dispersion ratio was the smallest when △t was 790 days.

### 4.2. Analysis of the Failure Number of Prediction Results

The distribution parameters were calculated with failure from the data from August 2017 to December 2018, August 2017 to December 2019, and August 2017 to December 2020, respectively. The calculated results are shown in Table 4.

### 4.3. Analysis of Relative Dispersion Degree of Prediction Results

The observation time was 487 days, 852 days, and 1217 days, respectively. The relationship between the prediction failure number H and the future prediction time interval △t was analyzed, as shown in Figure 3.

As shown in the figure above, H increased with the growth of △t, and went into a stable trend in the later stage gradually. With the continuous increase of △t, all the electricity meters would finally fail. On the whole, as the time △t increases, more meters need to be replaced, and we should increase the storage of meters accordingly.

According to the parameters of the calculation of the data in three observations, the relationship between the prediction failure number H in the next 365 days and the observation time $t_{s}$ was analyzed, as shown in the following figure.

As shown in Figure 4, H in the next 365 days decreased with the increase in $t_{s}$, and H predicted by the observed time of the previous 487 days was a temporary increase by the time of $t_{s}$, and then decreased with the rise of $t_{s}$, and the decline was faster than the other two. On the whole, we can see that the number of electricity meters to be replaced in the next year decreased with the increase in the observation time.

The observation time was 487 days, 852 days, and 1217 days, respectively. The relationship between the number of the predicted failure in the next 365 days and the total batch quantity N was analyzed, as shown in Figure 5.

The figure above shows that H in the next 365 days increased with the increase in N when $t_{s}$ and △t were unchanged, and H in the prediction of the data in the longer observed period increased more slowly with the rise in N. Therefore, as the number of meters installed in batches increases, the number of meters to be replaced in the next year will also increase, and we will need to reserve more meters accordingly.

The relationship between the relative dispersion ratio E and the future prediction time interval △t was analyzed, as shown in Figure 6.

As shown in the figure above, E decreased with △t and finally exhibited a stable trend. It can be seen that the value of E was first close to 1 and then away from 1 with the increase in △t, which indicated that the dispersion degree of the number of meter failure H decreased first and then increased.

The observation time was 487 days, 852 days, and 1217 days, respectively. The relationship between the relative dispersion ratio E in the next 365 days and the total quantity N was analyzed, as shown in Figure 7.

As shown in the figure above, E decreased with the increase of N when the time of $t_{s}$ and △t were all unchanged, and finally exhibited a stable trend. It can be seen that the value of dispersion ratio E first approached 1 and then moved away from 1 with the increase in the total number of electricity meters in batches, which indicated that the dispersion degree of the number of electricity meter failure decreased first and then increased.

### 4.4. Bayesian Fault Point Estimation and Interval Estimation Case Analysis

After multisource data fusion, the reliability life was 2920 days to 5840 days when the reliability R was 0.9. Since the data used for Bayesian prediction in this chapter were the field data from August 2017 to the end of 2019 in Table 1, the parameter $m_{b}$ was 0.91697 calculated from August 2017 to the end of 2019 without previous information.

The previous distribution parameters were obtained according to Equations (25) and (29): a = 95.17269, b = 1778004.98, and ${\hat{\lambda }}_{b}$ = 0.0000728898. The corresponding Bayes point estimates and interval estimates of failure number were obtained according to Equations (31), (32) and (34), as shown in Table 5 and Table 6 below.

After verification, the Bayesian prediction result of the number of failures in this example was slightly less than the actual number of on-site failures.

**Table 6 sensors-22-09804-t006:** Bayesian interval prediction and relative dispersion ratio.

△t	Bayes Bilateral Interval	$E_{b}$
365	[7.496, 19.828]	1.589
730	[17.180, 36.204]	1.249
790	[18.765, 38.820]	1.222

where △t is the future time interval, and the unit is days.

In this example, the Bayesian interval estimation results were consistent with the prediction results without previous information fusion.

The prediction results of the Bayes method and those without fusion of previous information change over time, as shown in the following figures (Figure 8, Figure 9, Figure 10 and Figure 11).

Above, the prediction results of H were increased with the increase in △t, and the failure number prediction results of the Bayes failure were smaller than those that do not merge the prior information. As △t was increasing, the final two kinds of failure number were expected to be close to the number of batches. Based on the Bayesian prediction results, the number of meters to be replaced in the predicted time interval △t was relatively small. Corresponding to the Bayesian prediction case, we only need to reserve fewer meters.

Based on the data of the previous 852 days of observation time, the relationship between the prediction failure number H of the next 365 days and the observation time $t_{s}$ was analyzed by using no prior information fusion and the Bayes method, as shown in Figure 9.

**Figure 9 sensors-22-09804-f009:**
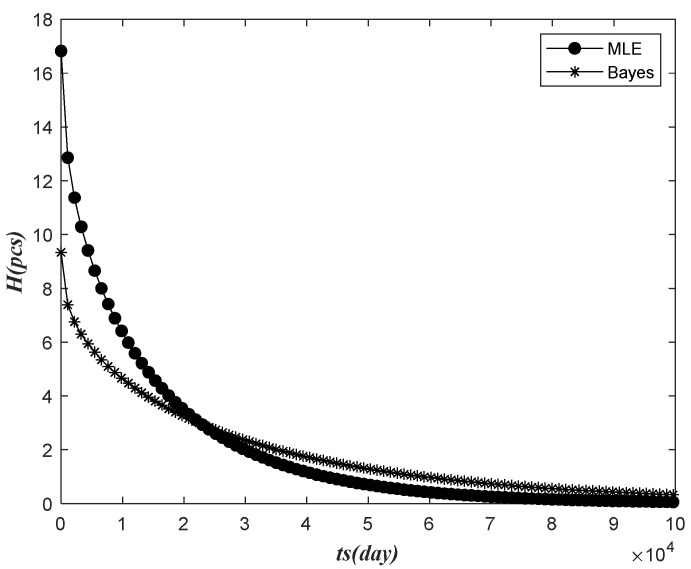
The relationship between Bayesian and classical prediction of failure times and the observation time variation.

Above, according to the data in the previous 852 days of observation time, H predicted by the two methods decreased with the increase in $t_{s}$ when the prediction period △t was the coming 365 days. The expected number of failures H by the Bayesian method decreased more slowly with the increase in $t_{s}$. In the end, H indicated by the two methods was stable with increasing observation time. With the rise in observation time, compared with the number of electricity meters to be replaced in the next year without prior information prediction, the number of electricity meters to be replaced in the next year under Bayesian prediction is first small and then large.

Based on the data of the previous 852 days of observation time, the relationship between the prediction failure number H of failures of the next 365 days and the total batch quantity N were analyzed using no prior information fusion and the Bayes method, as shown in Figure 10.

**Figure 10 sensors-22-09804-f010:**
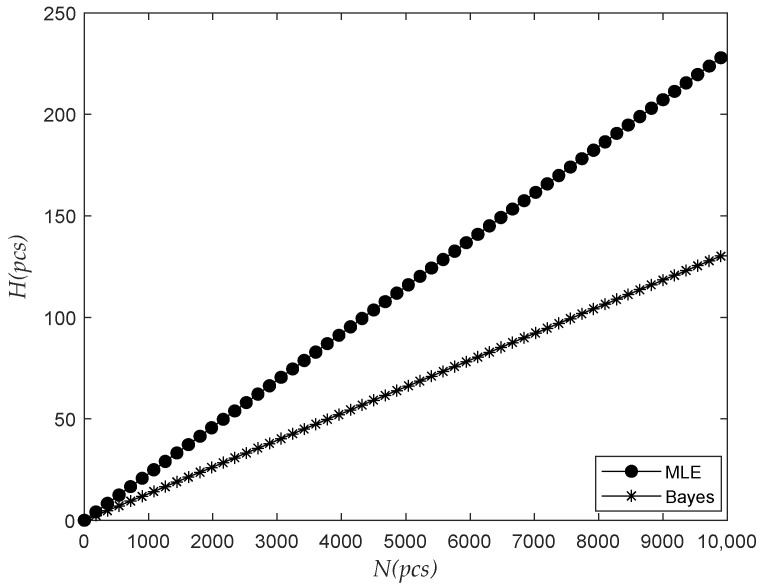
The relationship between Bayesian and classical prediction of failure times and the total quantity variation.

As seen from the above figure, H in the next 365 days increased with the increase in the number of batches when the time $t_{s}$ and the future prediction time interval △t were all unchanged, and H predicted by the Bayesian method increased more slowly with the increase in N. As the total number of meters installed in the same batch increases, the number of meters that will need to be replaced in the next year under the Bayesian prediction is less than that under the prediction without prior information.

Based on the data of the previous 852 days’ observation time, the relationship between the relative dispersion ratio E of the next 365 days and the prediction time interval △t was analyzed using no prior information fusion and the Bayes method, as shown in Figure 11.

**Figure 11 sensors-22-09804-f011:**
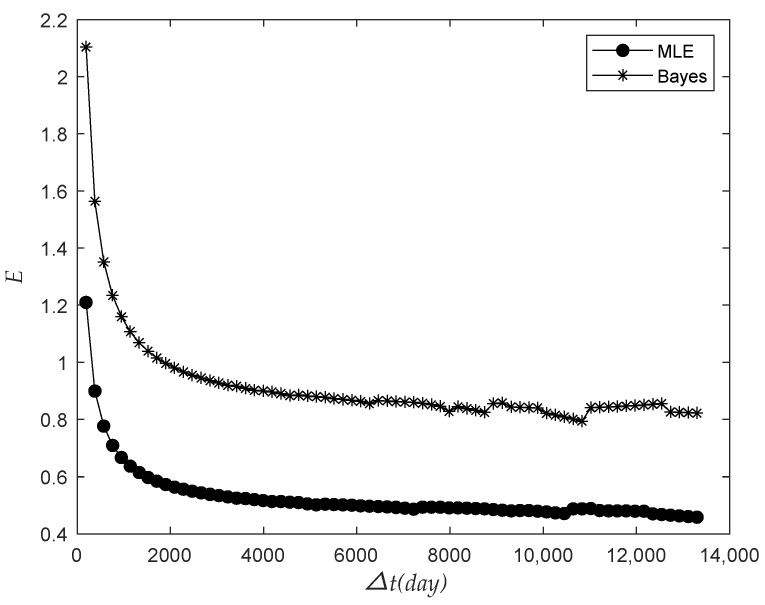
Bayesian and classical methods predict the relative dispersion of results with future time intervals.

As seen from the above figure, E decreased with the increase in △t and finally exhibited a stable trend, and the Bayesian dispersion ratio prediction result was more significant than the dispersion ratio prediction result without prior information. On the whole, it can be seen that the dispersion ratio of Bayesian prediction results was closer to 1, which indicated that the dispersion degree of Bayesian prediction results was lower.

Based on the data of the previous 852 days of observation time, the relationship between the relative dispersion ratio E of the next 365 days and the total batch quantity N was analyzed using no prior information fusion and the Bayes method, as shown in the following figure.

As seen from Figure 12, the Bayes prediction dispersion ratio curve was initially higher than the dispersion ratio curve without prior information, and then slowly approached consistency. The value of E calculated by the two methods decreased with the increase in N when the time of observation and the prediction time interval were all unchanged, and finally exhibited a stable trend. On the whole, it can be seen that the dispersion degree of Bayesian prediction results was consistent with that of prediction results without prior information.

It can be seen from the above figures that the Bayes prediction of the number of electric meter faults after information fusion was consistent with that without previous information fusion. In the prediction of the relative dispersion ratio, the relative dispersion degree of Bayesian prediction was lower and the effect was better.

### 4.5. Replacement Strategies for Smart Electricity Meters

There were 578 million installed in August 2017, and 56 million had broken down by the end of 2021. According to Equations (37), (39) and (41), the cumulated failure number of meters in the next year $r_{YF}$**_,_** the total failed meters in the next year $r_{Accu}$, and the proportion of failed meters to all meters $R_{F/ALL}$ was predicted. According to Equations (38), (40) and (42), the corresponding Bayesian estimation can be obtained, and the prediction results are shown in Table 7.

The threshold $R_{D,rotate}$ was set to 0.20, and the specified rotation time $T_{D,rotate}$ was eight years. According to the results in Table 5, the proportion of failed meters to all meters $R_{F/ALL}$ was 0.11486, and the corresponding Bayes estimate was 0.109395, which was less than $R_{D,rotate}$ = 0.20. The operation time of the meters was less than eight years. Therefore, the smart electricity meters did not need to be rotated.

## 5. Conclusions

A failure number prediction method for smart electricity meters based on Weibull distribution and odds ratio was proposed, and then strategies for electricity meter replacement and reserve were developed. The odds ratio was used to compute the distribution of failure number of meters, then the confidence intervals of distribution parameters and failure number in the future time interval were provided. Combined with the Bayes method and previous information, the number of meter failures was predicted. For the sake of application, replacement strategies, including replacement quantity, rotation time, and rotation criteria were developed, and a procedure for meter replacement was given. A case study validated the proposed methods. After verification, combined with field data, the method in this paper is effective and valuable for solving related problems. The data sources used in this paper are reliable and the method is novel. At the same time, the research results of this paper have been recognized bypartners, so the possibility of implementing the solution is beyond doubt. Next, we will continue to study methods predicting the number of failures based on historical data in the following year. We will explore ways to solve the problem of prediction and rotation of the failure number of smart electricity meters, and discuss the advantages and disadvantages of various methods.

## Figures and Tables

**Figure 1 sensors-22-09804-f001:**
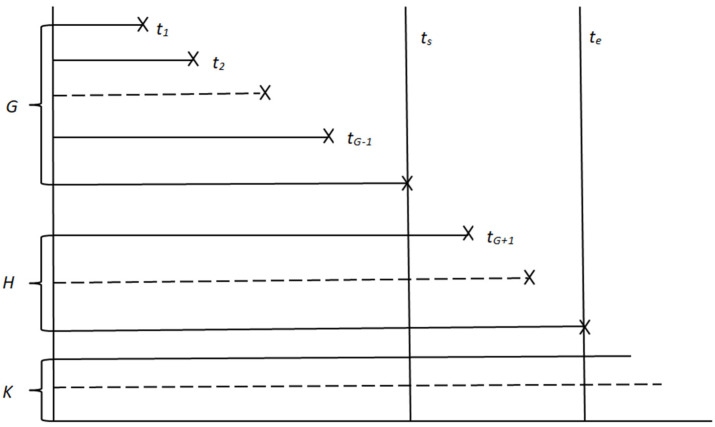
Prediction of the number of timing truncation failures.

**Figure 2 sensors-22-09804-f002:**
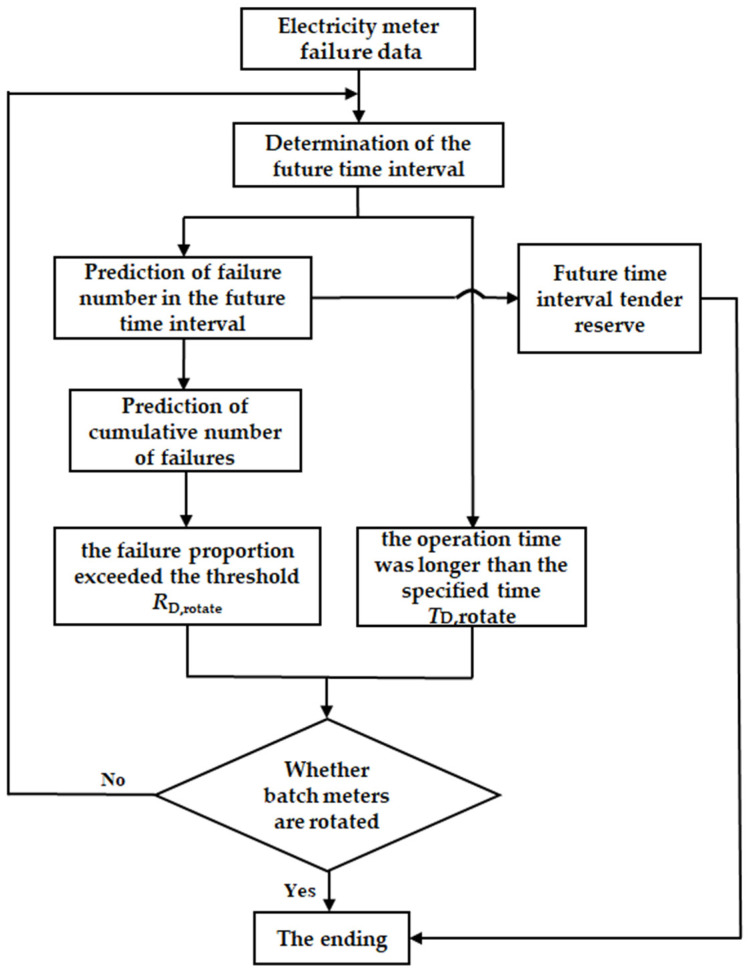
Decision-making flowchart of electricity meter reserve quantity.

**Figure 3 sensors-22-09804-f003:**
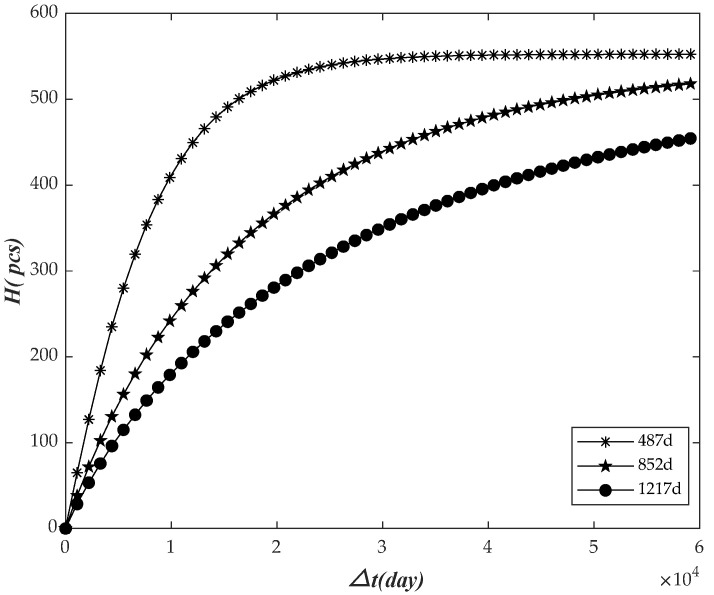
The relationship between the number of failures and the future interval.

**Figure 4 sensors-22-09804-f004:**
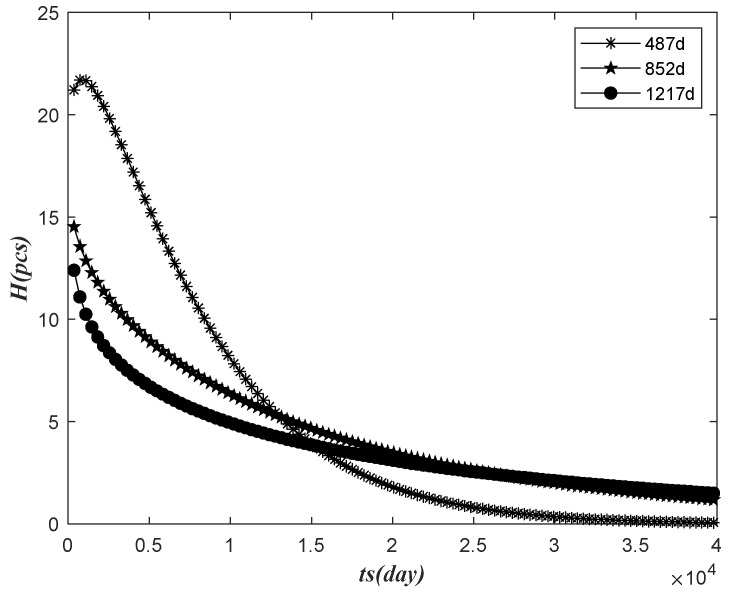
The relationship between the number of failures and the observation time.

**Figure 5 sensors-22-09804-f005:**
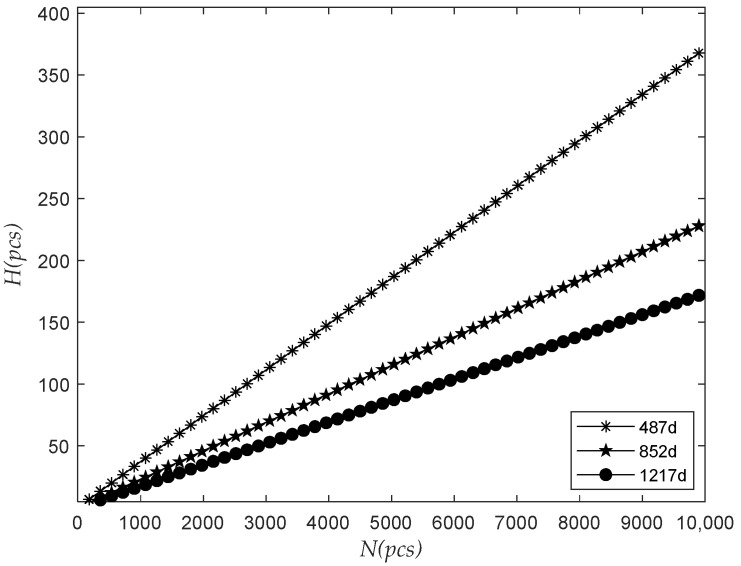
The relationship between the number of failures and the total batch quantity.

**Figure 6 sensors-22-09804-f006:**
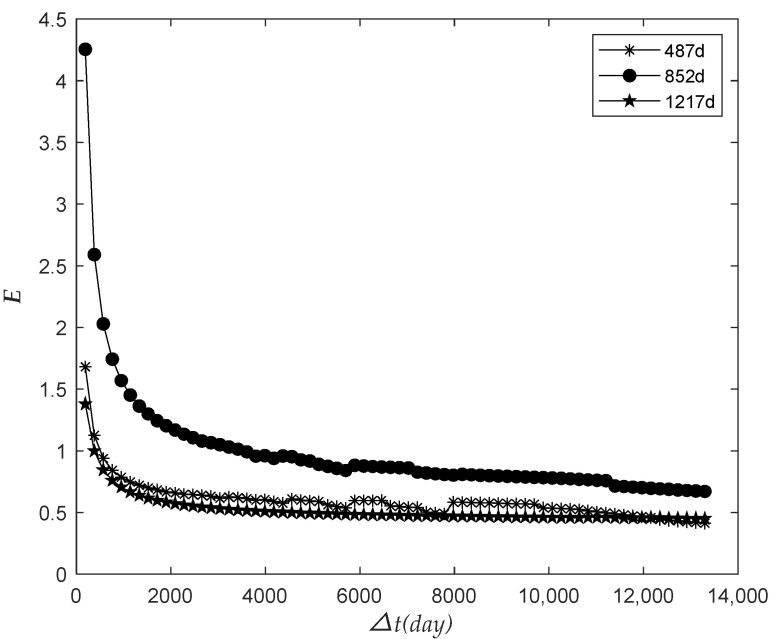
The relationship between the relative dispersion ratio and the future interval.

**Figure 7 sensors-22-09804-f007:**
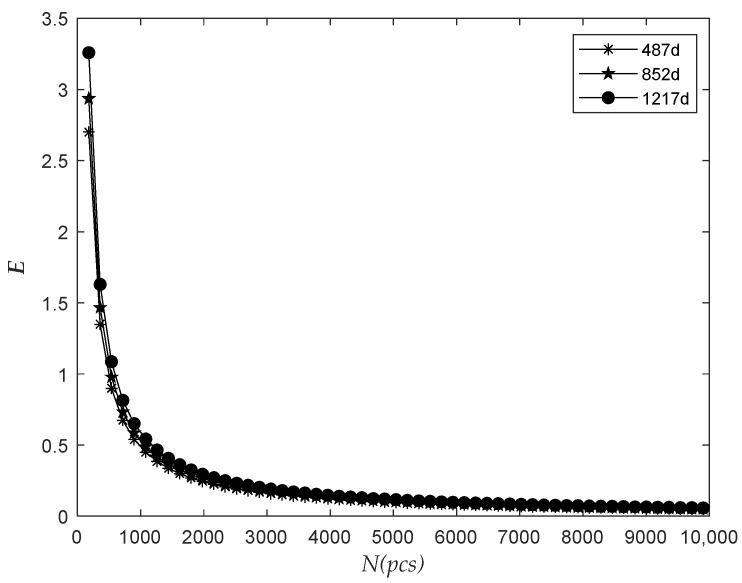
The relationship between the relative dispersion ratio and the total batch quantity.

**Figure 8 sensors-22-09804-f008:**
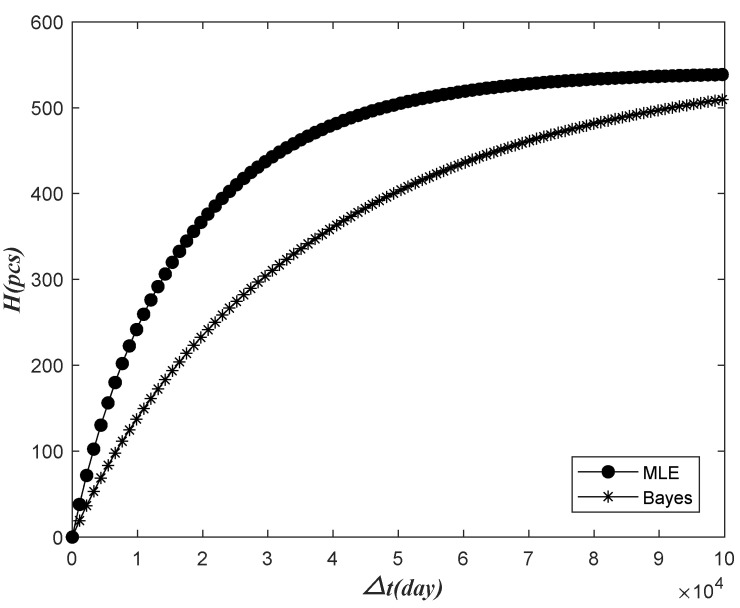
The relationship between Bayesian and classical prediction of failure times and future interval variation.

**Figure 12 sensors-22-09804-f012:**
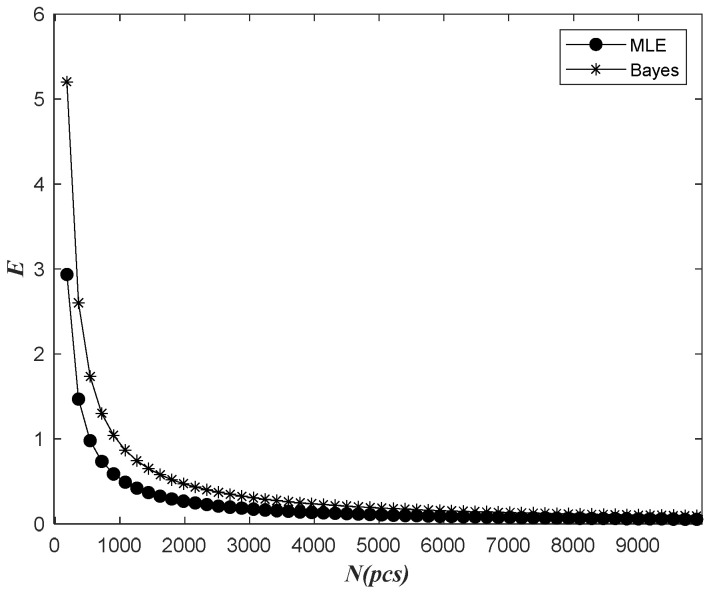
Bayesian and classical methods predict the relative dispersion of results with the total quantity variation.

**Table 1 sensors-22-09804-t001:** Fault data of electricity meters from August 2017 to December 2019.

$n\left(t\right)$	t	$\begin{array}{c} N-n\left(t\right) \end{array}$	$\begin{array}{c} F\left(t\right) \end{array}\;$	$n\left(t\right)\;$	t	$\begin{array}{c} N-n\left(t\right) \end{array}\;$	$\begin{array}{c} F\left(t\right) \end{array}\;$
1	11	577	0.001730	19	375	559	0.03287197
2	12	576	0.003460	20	382	558	0.03460207
3	57	575	0.005190	21	390	557	0.03633217
4	81	574	0.006920	22	403	556	0.03806228
5	140	573	0.008651	23	415	555	0.03979238
6	149	572	0.010381	24	416	554	0.04152249
7	154	571	0.012111	25	428	553	0.04325259
8	184	570	0.013841	26	492	552	0.04498269
9	200	569	0.015571	27	607	551	0.04671280
10	211	568	0.017301	28	622	550	0.04844291
11	215	567	0.019031	29	647	549	0.05017301
12	245	566	0.020761	30	667	548	0.05190310
13	253	565	0.022491	31	681	547	0.05363322
14	253	564	0.024221	32	738	546	0.05536332
15	267	563	0.025952	33	756	545	0.05709343
16	336	562	0.027681	34	782	544	0.05882353
17	341	561	0.029411	35	827	543	0.06055363
18	353	560	0.031142				

**Table 2 sensors-22-09804-t002:** Estimation of distribution parameters and number of failures.

η=16,995.978		m=0.91697	
△t	365	730	790
h	0.023	0.0449	0.0475
H	13.310	25.965	27.783

**Table 4 sensors-22-09804-t004:** Estimation of distribution parameters and number of failures.

Time	August 2017–Decemeber 2018	August 2017–Decemeber 2019	August 2017–Decemeber 2020
m	1.11963	0.91697	0.82748
η	7689.025	16,995.97	26,513.455

**Table 5 sensors-22-09804-t005:** Bayesian estimation of failure rate and number of failures.

${\hat{\lambda }}_{b}=0.0000728898$		$m_{b}=0.91697$	
△t	365	730	790
$h_{b}$	0.01342	0.02634	0.02843
$H_{b}$	7.75923	15.227	16.430

**Table 7 sensors-22-09804-t007:** Prediction results of failure number and failure ratio by Bayesian and classical methods.

The Next Year		$r_{\mathrm{YF}}$	$r_{\mathrm{Accu}}$	$R_{F/\mathrm{ALL}}$
2022	Classical	10.39	66.39	0.11486
	Bayesian	7.23	63.23	0.109395

## Data Availability

Not applicable.

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
