# Peer review of "Failure Prediction and Replacement Strategies for Smart Electricity Meters Based on Field Failure Observation†"

_sensors, 2022, doi:10.3390/s22249804_

Round 1

Reviewer 1 Report

a) What are the parameters to select a specified time interval? (sec.-2.1)

b) Technical explanation of the paper is very weak; please important sec. 2 & 3.

c) In the flow chart presented in Fig. 2, How ‘Instantaneous failure rate prediction’ and ‘Accumulated failure rate proportion prediction’ are related, and how do they check the condition of ‘batch meter are rotated’?

d) More information should be there about equations used in different sections.

e) Conclusion does not provide information about the future scope of work.

f) Result and discussion section should be explored.

g) Add nomenclature/abbreviation.

h) Paper formatting should be improved.

i) Some basic grammatical mistakes should be corrected.

Author Response

Thank you for your comments on our article. According to your suggestions, We have carefully studied the comments and have revised the manuscript.
Please see the attachment.

Reviewer 2 Report

Dear Authors,

I have some comments on your article:

1. Literature should be checked if there are no newer items. Especially from the last 18 months.

2. All indexes in symbols in text and equations should be checked carefully.

3. It would be good to provide some information about electricity meters in the article.

4. In the Conclusions section, please write something more about the possibility of implementing the solution.

5. At what minimum observation period failures can be detected?

Author Response

(The authors gave the same response as above.)

Round 2

Reviewer 2 Report

Dear Authors,

Thank you very much for introducing changes that have improved the quality of the article. I have no more comments.

Best regards